# Probabilistic Feature Imputation and Uncertainty-Aware Multimodal Federated Aggregation

**Nafis Fuad Shahid**[1]   NAFISFUAD21@IUT-DHAKA.EDU

**Maroof Ahmed**[*1]   MAROOFAHMED@IUT-DHAKA.EDU

**Md Akib Haider**[*1]   AKIBHAIDER@IUT-DHAKA.EDU

**Saidur Rahman Sagor**[1]   SAIDURRAHMAN@IUT-DHAKA.EDU

**Aashnan Rahman**[1]   AASHNANRAHMAN@IUT-DHAKA.EDU

**Md Azam Hossain**[1]   AZAM@IUT-DHAKA.EDU

[1] *Department of Computer Science and Engineering, Islamic University and Technology (IUT), Gazipur, Bangladesh*

**Editors:** Accepted for publication at MIDL 2026

## Abstract

Multimodal federated learning enables privacy-preserving collaborative model training healthcare applications. However, a fundamental challenge arises from modality heterogeneity: many clinical sites possess only a subset of modalities due to resource constraints or workflow variations. Existing approaches address this through feature imputation networks that synthesize missing modality representations, yet these methods produce point estimates without reliability measures, forcing downstream classifiers to treat all imputed features as equally trustworthy. In safety-critical medical applications, this limitation poses significant risks. We propose the Probabilistic Feature Imputation Network (P-FIN), which outputs calibrated uncertainty estimates alongside imputed features. This uncertainty is leveraged at two levels: (1) locally, through sigmoid gating that attenuates unreliable feature dimensions before classification, and (2) globally, through Fed-UQ-Avg, an aggregation strategy that prioritizes updates from clients with reliable imputation. Experiments on federated chest X-ray classification using CheXpert, NIH Open-I, and PadChest demonstrate consistent improvements over deterministic baselines, with +5.36% AUC gain in the most challenging configuration. Code implementation is available at https://github.com/NafisFuadShahid/PFIN-UQAVG

**Keywords:** Federated Learning, Multimodal Learning, Uncertainty Quantification, Feature Imputation, Medical Imaging

## 1. Introduction

In modern healthcare, relying on a single source of information to formulate a diagnosis is insufficient and potentially unsafe (Teoh et al., 2024). Instead, diverse data types, such as radiological images, clinical history, and textual reports are synthesized. Research consistently shows that multimodal models, which learn from both images and text, significantly outperform models that rely on images alone (Huang et al., 2020; Acosta et al., 2022). For instance, in chest X-ray analysis, combining the visual scan with the radiologist's textual report enables the system to capture complex medical conditions with greater accuracy (Zhang et al., 2022; Boecking et al., 2022).

---

* Contributed equally

SHAHID◎ AHMED◎ HAIDER◎ SAGOR◎ RAHMAN◎ HOSSAIN◎

However, the development of these collaborative AI models is constrained by strict privacy regulations. Laws such as HIPAA and GDPR prohibit the centralization of patient data across institutions (Price and Cohen, 2019). To address this, Federated Learning (FL) was introduced (McMahan et al., 2017) to collaboratively train a shared model without transferring patient data off-site among hospitals. Data remains private locally, and only model updates are shared (Rieke et al., 2020; Sheller et al., 2020).

While Federated Learning resolves privacy concerns, it encounters a practical challenge known as modality heterogeneity. In real-world deployments, resources vary significantly across institutions. Large academic medical centers often possess complete datasets comprising both X-rays and detailed reports. Conversely, smaller clinics or rural hospitals may only have access to X-ray images, lacking the infrastructure to provide structured text data (Rajpurkar et al., 2017; Warnat-Herresthal et al., 2021). This disparity creates a network where some participants possess complete multimodal data, while others hold incomplete unimodal data.

To mitigate this missing data problem, early research proposed feature imputation, which trains the model to synthesize the missing modality. For example, if a clinic provides only an image, the model predicts the corresponding text report based on visual patterns (Ngiam et al., 2011). Recent approaches, such as SMIL (Ma et al., 2021), have refined this process for complex datasets. However, a critical flaw persists in these methods: they are deterministic. A deterministic model produces a single confident prediction even when guessing, potentially hallucinating a text description for an ambiguous image with high confidence. In safety-critical medical domains, such silent failures can lead to erroneous diagnoses (Jungo and Reyes, 2019; Nair et al., 2020).

We argue that AI systems must be transparent about their limitations. Regulatory bodies, including the U.S. Food and Drug Administration (FDA), explicitly state the need to develop methods to quantify uncertainty and convey it in the device output to users (U.S. Food and Drug Administration, 2024). This concept, known as Uncertainty Quantification (UQ), is essential for safety. Foundational work by Kendall and Gal (2017) established methods to measure this uncertainty in deep learning. By integrating UQ, a model can report high uncertainty when input data is ambiguous, thereby warning clinicians not to trust the synthetic features.

In this paper, we propose the **Probabilistic Feature Imputation Network (P-FIN)**. Instead of deterministically estimating missing features, our model outputs a distribution parameterized by a mean value and a variance score. We train this network using a specialized loss function called $\beta$-NLL (Seitzer et al., 2022), which prevents the model from minimizing loss by simply predicting infinite uncertainty. We leverage this uncertainty in two distinct ways. Locally, the variance acts as a gate to suppress unreliable features before fusion. Globally, we introduce **Fed-UQ-Avg**, a novel aggregation method that prioritizes updates from hospitals with confident, high-quality data over those with high uncertainty. Our experiments on diverse chest X-ray datasets (Irvin et al., 2019; Bustos et al., 2020) demonstrate that this approach is significantly more robust than previous methods.

## 2. Related Work

### 2.1. Missing Modalities in Federated Learning

Feature imputation networks have emerged as a solution to missing modality challenges by learning cross-modal mapping through various architectural approaches. Early work by Ngiam et al. (Ngiam et al., 2011) introduced deterministic autoencoders for synthesizing cross modality representations. Ma et al. (Ma et al., 2021) proposed SMIL, a Bayesian meta-learning approach for severely missing modalities while Kaissis et al. (Kaissis et al., 2020) and Warnat-Herresthal et al. (Warnat-Herresthal et al., 2021) adapted deterministic feature imputation for federated medical imaging. However, existing methods produce point estimates without reliability measures, precluding uncertainty-aware downstream processing.

### 2.2. Uncertainty Quantification in Medical Imaging

Uncertainty estimation is critical for safety-critical medical applications, enabling systems to flag unreliable predictions for human review. Kendall and Gal (Kendall and Gal, 2017) distinguished aleatoric uncertainty (inherent data noise) from epistemic uncertainty (model ignorance), with heteroscedastic aleatoric modeling being particularly relevant for input-dependent reliability assessment. Bayesian approaches including Monte Carlo Dropout (Gal and Ghahramani, 2016) and variational inference have been applied to medical image segmentation (Jungo and Reyes, 2019; Nair et al., 2020) and selective prediction (Laves et al., 2020; Geifman and El-Yaniv, 2019). However, while these works primarily focus on single-modal centralized settings, uncertainty quantification remains underexplored in multimodal federated settings, where both feature imputation reliability and client contribution quality must be assessed.We address this gap by introducing probabilistic feature imputation that explicitly models uncertainty during cross-modal synthesis, and leverage these uncertainty estimates both locally for feature gating and globally for uncertainty-aware federated aggregation.

### 2.3. Calibration and Loss Functions

Training neural networks to predict calibrated uncertainty is non-trivial. Guo et al. (Guo et al., 2017) demonstrated that modern deep networks are often miscalibrated, producing overconfident predictions. Standard Gaussian negative log-likelihood (NLL) training can lead to "variance explosion," where models predict infinite uncertainty to minimize loss without learning meaningful representations. Recent approaches address this through various regularization strategies: Lakshminarayanan et al. (Lakshminarayanan et al., 2017) proposed deep ensembles, Laves et al. (Laves et al., 2021) introduced $\sigma$-scaling for recalibration, and Seitzer et al. (Seitzer et al., 2022) developed $\beta$-NLL loss using stop-gradient operations. We adopt $\beta$-NLL for training our probabilistic imputation network due to its effectiveness in preventing variance collapse.

### 2.4. Federated Aggregation Strategies

FedAvg (McMahan et al., 2017) remains the dominant aggregation strategy, weighting client contributions proportionally to local dataset size. FedProx (Li et al., 2020) addresses

statistical heterogeneity through proximal regularization, while other works have explored adaptive weighting based on gradient similarity or loss values (Sheller et al., 2020; Dayan et al., 2021). However, existing aggregation strategies do not account for feature reliability in missing modality scenarios. A client with many samples but poor imputation quality can degrade the global model through noisy gradient contributions. Our Fed-UQ-Avg explicitly incorporates imputation confidence, complementing data-based weighting with quality-aware adjustments.

## 3. Methodology

We consider a federated learning setting with $K$ clients. A subset $\mathcal{C}_m$ (multimodal clients) possesses paired chest X-rays $x^I$ and radiology reports $x^T$, while the remaining clients $\mathcal{C}_u$ (unimodal clients) have only images. Our objective is to enable effective multimodal learning across all clients by providing reliable imputation for missing modalities.

### 3.1. Feature Encoders

The image encoder $f_I$ employs a ResNet-50 backbone (He et al., 2016) pretrained on ImageNet, with the final classification layer replaced by a linear projection:

$$z^I = \frac{W_I \cdot \text{ResNet}(x^I)}{\|W_I \cdot \text{ResNet}(x^I)\|_2} \in \mathbb{R}^{256} \tag{1}$$

where $W_I \in \mathbb{R}^{256 \times 2048}$ projects ResNet features to a 256-dimensional space, followed by L2 normalization. The text encoder $f_T$ uses BERT-base-uncased (Devlin et al., 2019), extracting the [CLS] token representation with analogous projection:

$$z^T = \frac{W_T \cdot \text{BERT}(x^T)_{[\text{CLS}]}}{\|W_T \cdot \text{BERT}(x^T)_{[\text{CLS}]}\|_2} \in \mathbb{R}^{256} \tag{2}$$

where $W_T \in \mathbb{R}^{256 \times 768}$. Both encoders output L2-normalized features to ensure compatible representation spaces.

### 3.2. Probabilistic Feature Imputation Network (P-FIN)

Unlike deterministic approaches that output a fixed vector $\hat{z}^T$, P-FIN models the conditional distribution $p(z^T|z^I)$ as a heteroscedastic Gaussian. The architecture comprises:

**Input Projection.** Image features are projected and reshaped:

$$h_0 = \text{GELU}(\text{LayerNorm}(\text{Linear}(z^I))) \in \mathbb{R}^{1 \times 256} \tag{3}$$

**Learnable Query Token.** A learnable parameter $q \in \mathbb{R}^{1 \times 256}$ is combined with $h_0$, forming the input sequence $[q; h_0] \in \mathbb{R}^{2 \times 256}$.

**Transformer Encoder.** A 2-layer Transformer encoder with 4 attention heads processes the sequence:

$$h_L = \text{TransformerEncoder}([q; h_0]) \in \mathbb{R}^{2 \times 256} \tag{4}$$

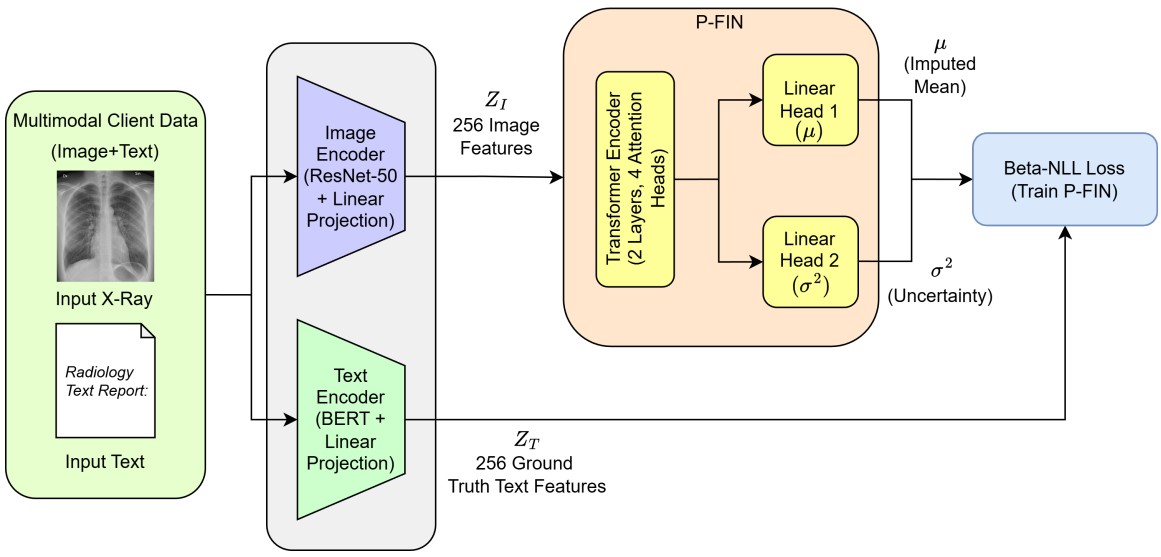

Figure 1: Overview of Stage 1: P-FIN Training. The architecture leverages a Transformer encoder to map image features to text embedding distributions, trained via $\beta$-NLL loss for calibrated uncertainty.

The query output $h_L[0]$ contains the aggregated cross-modal information.

**Dual Output Heads.** Two separate MLPs predict the mean and variance:

$$\mu = \text{MLP}_\mu(h_L[0]) \in \mathbb{R}^{256} \tag{5}$$

$$\sigma^2 = \text{MLP}_\sigma(h_L[0]) \in \mathbb{R}^{256} \tag{6}$$

The variance $\sigma^2$ represents the per-dimension **uncertainty**, which is directly output by the network and used for both gating and aggregation throughout this work.

### 3.3. Calibrated Training via $\beta$-NLL

Standard Gaussian NLL allows models to minimize loss by predicting large variances without learning meaningful features. We train P-FIN using the $\beta$-NLL loss (Seitzer et al., 2022), which applies a stop-gradient to prevent this shortcut:

$$\mathcal{L}_{\beta\text{-NLL}} = \frac{1}{d} \sum_{j=1}^{d} \text{SG}(\sigma_j^{2\beta}) \left( \frac{1}{2} \log \sigma_j^2 + \frac{(z_j^T - \mu_j)^2}{2\sigma_j^2} \right) \tag{7}$$

where $\text{SG}(\cdot)$ denotes the stop-gradient operator and $d = 256$. Setting $\beta = 0.5$ balances calibration with reconstruction quality, forcing the model to reduce prediction error rather than inflate uncertainty.

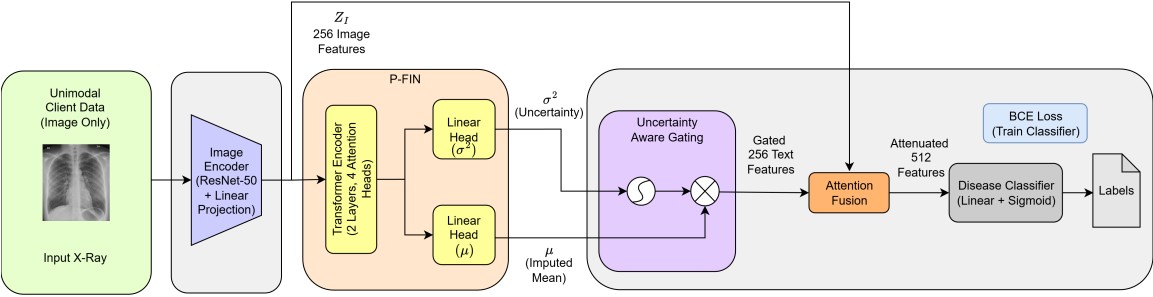

Figure 2: Overview of Stage 2: P-FIN Inference. The gating mechanism $g$ attenuates unreliable dimensions of the imputed feature vector $\mu$ based on uncertainty $\sigma^2$ before attention-guided fusion and classification.

### 3.4. Local Uncertainty-Aware Fusion

On unimodal clients, directly using imputed features $\mu$ can propagate errors when imputation is unreliable. We introduce uncertainty-aware fusion that combines gating mechanisms (Arevalo et al., 2017) with cross-modal attention (Lee et al., 2018).

**Uncertainty Gating.** We compute a gate from the predicted variance:

$$g = \mathrm{sigmoid}(-\log \sigma^2) \in \mathbb{R}^{256} \tag{8}$$

When uncertainty $\sigma^2$ is high, $\log \sigma^2$ becomes large, making $-\log \sigma^2$ strongly negative, so the gate approaches zero ($g \to 0$), suppressing unreliable features. Conversely, when uncertainty is low, the gate remains close to one.

**Cross-Modal Attention.** Image and gated text features attend to each other bidirectionally:

$$\hat{z}^I = \mathrm{LN}(z^I + \mathrm{MHA}(Q{=}z^I, K{=}g \odot \mu, V{=}g \odot \mu)) \tag{9}$$

$$\hat{z}^T = \mathrm{LN}(g \odot \mu + \mathrm{MHA}(Q{=}g \odot \mu, K{=}z^I, V{=}z^I)) \tag{10}$$

where LN denotes layer normalization and MHA is multi-head attention (Vaswani et al., 2017).

**Fusion.** The attended features are concatenated and projected:

$$z_{\mathrm{fused}} = \mathrm{Linear}([\hat{z}^I; \hat{z}^T]) \in \mathbb{R}^{256} \tag{11}$$

This attention-guided fusion allows the model to dynamically weight contributions from each modality based on learned relevance, while the uncertainty gating ensures unreliable imputations are suppressed before fusion.

### 3.5. Global Aggregation: Fed-UQ-Avg

Standard FedAvg weights client contributions by dataset size alone, creating vulnerability when unimodal clients produce poor imputations. Fed-UQ-Avg incorporates imputation quality through a two-component weighting scheme.

**Data Weight.** Standard size-proportional weight:

$$w_{\text{data}}^{(k)} = \frac{n_k}{\sum_{j=1}^{K} n_j} \tag{12}$$

**Confidence Weight.** We compute confidence using a temperature-scaled exponential of the negative mean uncertainty:

$$\text{conf}_k = \exp\left(-\frac{\bar{\sigma}_k^2}{T}\right) \tag{13}$$

where $\bar{\sigma}_k^2$ is client $k$'s mean imputation uncertainty and $T$ is a temperature parameter. The normalized confidence weight is:

$$w_{\text{conf}}^{(k)} = \frac{\text{conf}_k}{\sum_{j=1}^{K} \text{conf}_j} \tag{14}$$

**Combined Weight.** The final aggregation weight blends both components:

$$W_k = (1 - \alpha)w_{\text{data}}^{(k)} + \alpha w_{\text{conf}}^{(k)} \tag{15}$$

We set $\alpha = 0.6$ and $T = 0.2$, prioritizing imputation reliability while accounting for data size. Clients with low uncertainty receive higher weights, while those with high uncertainty contribute less to the global model.

---

**Algorithm 1:** Fed-UQ-Avg

---

**Input:** Global Model $\boldsymbol{\theta}^0$, Temperature $T$, Balance $\alpha$, Clients $\mathcal{K}$
**Output:** Final Model $\boldsymbol{\theta}^R$
**for** $t \leftarrow 1$ **to** $R$ **do**
    Server broadcasts $\boldsymbol{\theta}^{t-1}$ to clients $\mathcal{K}$;
    *Parallel Client Training*
    **for** $k \in \mathcal{K}$ ***in parallel*** **do**
        **if** *Unimodal* **then** Impute missing modality with P-FIN, then train
        **else** Train with multimodal data
        Return $\boldsymbol{\theta}_k^t, \bar{\sigma}_k^2, n_k$
    **end**
    *Compute Weights and Aggregate*
    $N_{\text{tot}} \leftarrow \sum_{j \in \mathcal{K}} n_j$
    $Z_{\text{conf}} \leftarrow \sum_{j \in \mathcal{K}} \exp(-\bar{\sigma}_j^2/T)$
    $\boldsymbol{\theta}^t \leftarrow \mathbf{0}$
    **for** $k \in \mathcal{K}$ **do**
        $w_{\text{data}} \leftarrow n_k/N_{\text{tot}}$
        $w_{\text{conf}} \leftarrow \exp(-\bar{\sigma}_k^2/T)/Z_{\text{conf}}$
        $\lambda_k \leftarrow (1-\alpha)w_{\text{data}} + \alpha w_{\text{conf}}$
        $\boldsymbol{\theta}^t \leftarrow \boldsymbol{\theta}^t + \lambda_k \boldsymbol{\theta}_k^t$
    **end**
**end**
**return** $\boldsymbol{\theta}^R$

---

## 4. Experiments

### 4.1. Datasets

We use three publicly available chest X-ray datasets for our experiments. CheXpert (Irvin et al., 2019) contains 224,316 radiographs from 65,240 patients with 14 pathology labels extracted from reports. NIH Open-I (Wang et al., 2017) provides 7,470 images paired with 3,955 radiology reports, enabling multimodal learning with both visual and textual data. PadChest (Bustos et al., 2020) comprises 160,868 images from 67,000 patients, used for external validation across different institutions and patient populations.

### 4.2. Baselines

We compare against two standard heuristic approaches that serve as performance lower bounds, and a state-of-the-art deterministic imputation method representing the current standard in federated learning

**Standard Heuristics:** We employ two naive heuristics: **Zero-filling**, which replaces missing features with null vectors ($\hat{z}^T = \mathbf{0}$), and **Uniform-filling**, which substitutes the global mean embedding, representing a static baseline that ignores sample-specific visual context.

**FIN + FedAvg:** The approach proposed by (Poudel et al., 2025) uses a deterministic Transformer decoder to reconstruct bottleneck features via MSE loss, aggregated with standard FedAvg. Unlike our method, it produces point estimates without uncertainty quantification.

**P-FIN + FedAvg:** An ablation of our method utilizing standard FedAvg. This isolates the specific contribution of our uncertainty-aware global aggregation mechanism.

### 4.3. Setup and Implementation

We simulate federated environments using $K = 10$ clients distributed in three configurations of unimodal-to-multimodal ratios (8:2, 6:4, 4:6) to reflect varying data scarcity. Data were sourced from CheXpert (unimodal) and NIH Open-I (multimodal), distributed via a Dirichlet distribution ($\alpha_{\text{Dir}} = 0.5$) to ensure realistic non-IID label heterogeneity. PadChest was reserved for external validation.

P-FIN utilizes a ResNet-50 visual backbone and BERT-base textual backbone, both projected to 256 dimensions. The imputation module consists of a 2-layer Transformer with 4 attention heads and $d_{\text{model}} = 256$, trained with $\beta$-NLL ($\beta = 0.5$). For attention-guided fusion, we employ a single-head attention layer with 256-dimensional queries, keys, and values. Federated training spanned 20 communication rounds with 4 local epochs per round (batch size 32, Adam optimizer, learning rate $10^{-4}$). The Fed-UQ-Avg hyperparameters were set to $\alpha = 0.6$ and $T = 0.2$.

### 4.4. Results and Analysis

Table 1 summarizes classification performance (mean AUC across 14 classes) on held-out test data. P-FIN integrated with Fed-UQ-Avg demonstrated consistent superiority over all deterministic baselines across all configurations.

Table 1: Test AUC (%) across federated configurations with varying unimodal (I) to multimodal (M) client ratios. Best results in **bold**.

| Method | I:M = 8:2 | I:M = 6:4 | I:M = 4:6 |
|---|---|---|---|
| Zero-filling | $72.56 \pm 0.31$ | $75.61 \pm 0.33$ | $77.86 \pm 0.29$ |
| Uniform filling | $71.78 \pm 0.47$ | $74.18 \pm 0.53$ | $76.67 \pm 0.46$ |
| FIN + FedAvg | $77.40 \pm 0.57$ | $79.31 \pm 0.62$ | $81.28 \pm 0.83$ |
| P-FIN + FedAvg | $79.93 \pm 0.79$ | $81.72 \pm 0.37$ | $83.11 \pm 0.62$ |
| **Ours (P-FIN + Fed-UQ-Avg)** | $\mathbf{82.76} \pm 0.63$ | $\mathbf{84.14} \pm 0.49$ | $\mathbf{85.89} \pm 0.38$ |

In the most challenging 8:2 scenario, where 80% of clients lacked textual data, our approach yielded a 5.36% absolute improvement over standard deterministic imputation (FIN + FedAvg). This gain emphasizes the criticality of uncertainty-aware weighting when high-quality ground truth is scarce. Convergence analysis further revealed that mean imputation uncertainty on unimodal clients decreased progressively over communication rounds, validating that P-FIN successfully learned to approximate the missing modality distributions from the multimodal minority.

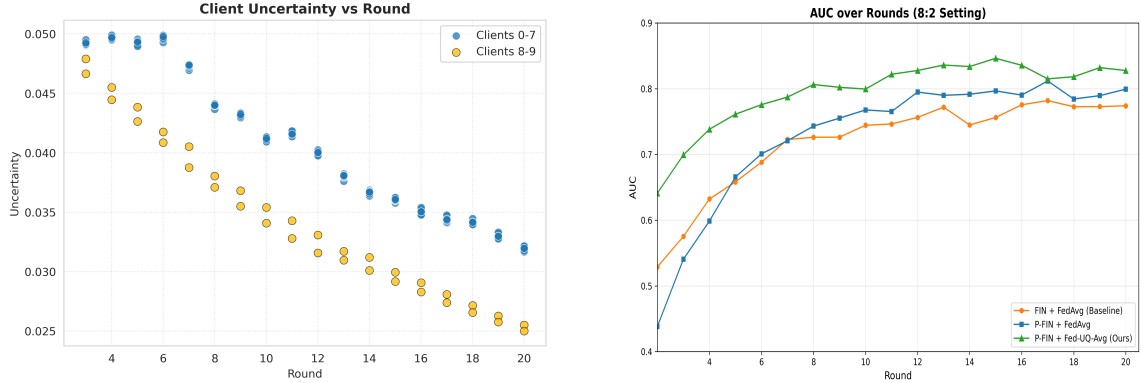

Figure 3: **(Left)** Evolution of uncertainty estimates for all clients (0–9). Unimodal clients (0–7) are shown in blue, while multimodal clients (8–9) are in orange. **(Right)** AUC progression per round.

The ablation study demonstrates the complementary value of both components: while P-FIN with standard FedAvg already improves over deterministic baselines by modeling imputation uncertainty, the addition of Fed-UQ-Avg yields further consistent gains of 2.83%, 2.42%, and 2.78% across the three heterogeneity configurations. This confirms that uncertainty-aware aggregation is essential for fully leveraging calibrated estimates, particularly in scenarios with high modality imbalance where unreliable client updates could otherwise dominate the global model.

The attention-guided fusion mechanism proved particularly beneficial compared to simple fixed-weight baselines. By allowing the model to dynamically weight the contribution of observed versus imputed features, the attention mechanism naturally leverages the gating signal to suppress unreliable imputations during feature aggregation.

### 4.5. Uncertainty Calibration Analysis

A primary claim of P-FIN is that predicted uncertainties are calibrated and meaningful. We validate this through three analyses.

We measure calibration using Expected Calibration Error (ECE) (Guo et al., 2017). This quantifies the discrepancy between expected and observed coverage. Figure 4(a) shows the reliability diagram for P-FIN. The close alignment between the observed coverage curve and the perfect calibration diagonal indicates dependable calibration (ECE = 0.0422).

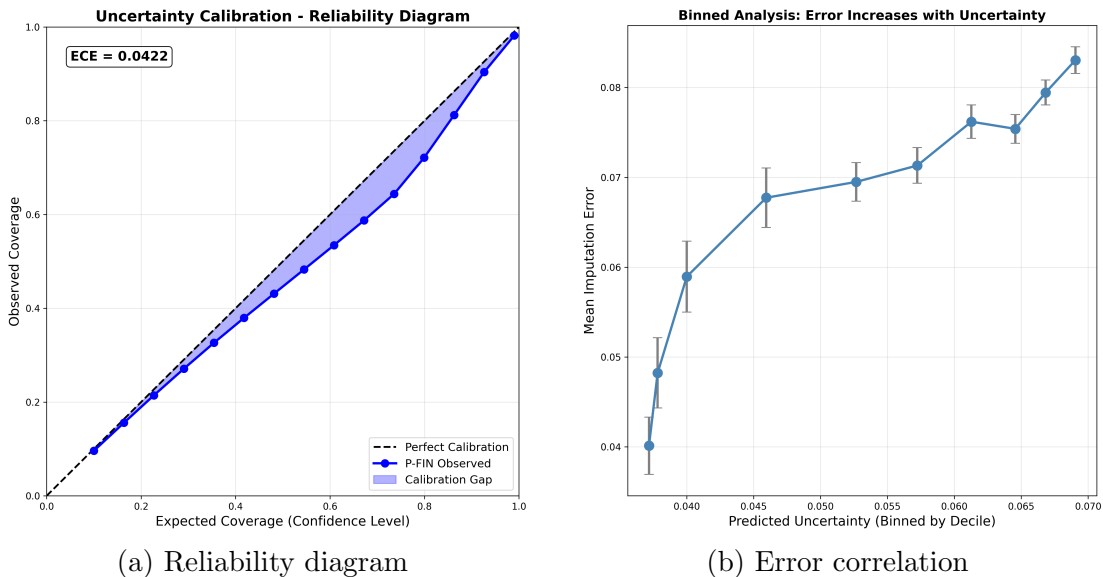

(a) Reliability diagram  (b) Error correlation

Figure 4: Uncertainty calibration analysis. (a) Reliability diagram showing dependable calibration with ECE = 0.0422. (b) Binned analysis demonstrating strong correlation between predicted uncertainty and reconstruction error.

We also analyzed the relationship between predicted uncertainty and actual imputation error to verify that uncertainty meaningfully reflects imputation quality. Figure 4(b) presents a binned analysis where samples are grouped into deciles by predicted uncertainty. The monotonic trend with mean imputation error increasing across uncertainty deciles confirms that high uncertainty reliably indicates low-quality imputations.

## 5. Discussion

The results substantiate the hypothesis that probabilistic modeling offers a robust defense against modality heterogeneity in federated learning. By explicitly quantifying imputation

**Qualitative Evaluation: Uncertainty Reflects Imputation Quality**

**LOW UNCERTAINTY → HIGH QUALITY IMPUTATION**

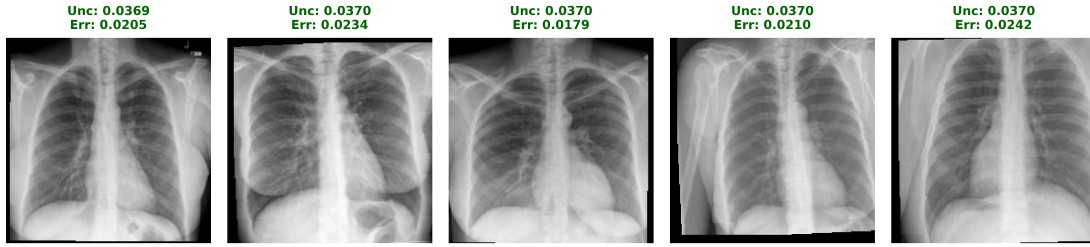

**HIGH UNCERTAINTY → LOW QUALITY IMPUTATION**

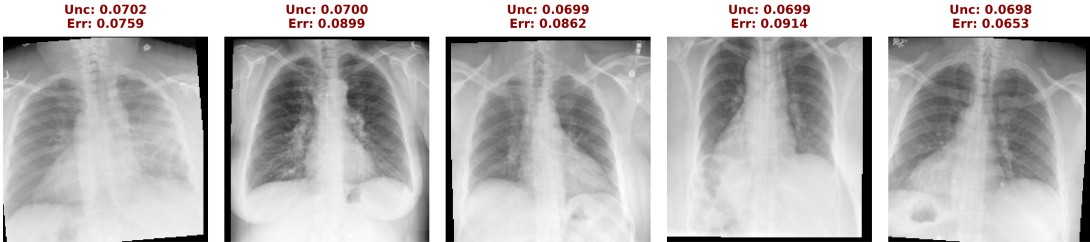

Figure 5: Qualitative Examples: Top row shows low-uncertainty cases with clear imaging; bottom row shows high-uncertainty cases with complex or ambiguous presentations.

uncertainty, P-FIN mitigates the risks of error propagation inherent in deterministic approaches. The dual-mechanism strategy, comprising local gating and global Fed-UQ-Avg, creates a synergistic effect: local gating prevents individual classifiers from overfitting to hallucinations, while global aggregation prevents the shared model from being corrupted by clients with poor imputation capabilities.

From a clinical perspective, P-FIN aligns with regulatory priorities for uncertainty quantification in AI-enabled medical devices (U.S. Food and Drug Administration, 2024). The ability to output a confidence metric alongside a prediction is a prerequisite for human-in-the-loop workflows, allowing radiologists to trust the model when uncertainty is low and scrutinize it when uncertainty is high. While our current implementation focuses on unidirectional image-to-text imputation, future work will explore bidirectional synthesis and the integration of epistemic uncertainty estimation to further enhance reliability in out-of-distribution scenarios.

## 6. Conclusion

We presented P-FIN, a probabilistic framework for handling missing modalities in federated medical imaging. By replacing point estimates with calibrated distributions, P-FIN enables uncertainty-aware local fusion and quality-weighted global aggregation. Our evaluation on chest X-ray classification confirms that this "learning to distrust" paradigm significantly outperforms deterministic alternatives, particularly in data-scarce environments. This work provides a foundation for more resilient and trustworthy multimodal federated learning systems in healthcare. Future work may explore more complex multimodal application scenarios as well as alternate aggregation mechanisms.

## Acknowledgments

We thank the contributors of the CheXpert, NIH Open-I, and PadChest datasets for making their data publicly available for research. Additionally, we wish to acknowledge Syed Rifat Raiyan and Reaz Hassan Joader, Department of Computer Science and Engineering, IUT, for their assistance in proofreading and offering a preliminary review of this manuscript

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
