# OpenReview forum: "Probabilistic Feature Imputation and Uncertainty-Aware Multimodal Federated Aggregation"
_MIDL.io/2026/Conference — MIDL 2026 Poster_

### Official Review · Reviewer_cxwa · 2026-01-08

**Confidence:** 3
**Preliminary Rating:** 2
**Final Rating:** 4

**Summary:**

The authors of this paper introduce the Probabilistic Feature Imputation Network which incorporates uncertainty quantification into feature imputation in case of missing modalities in multi-modal learning. It uses uncertainty locally as a gate to ensure sensibly imputed features and uses it globally to prioritise federated learning model updates that are based on confident, high-quality data

**Strengths:**

- Indirect ablation study (table 1) showing the benefit of both P-FIN (local use of uncertainty) and Fed-UQ-Avg (global use of uncertainty)
- Demonstrating that leveraging uncertainty directly improves performance

**Weaknesses:**

- Setting temperature parameter $T=1$ means no scaling was applied, how was $T$ chosen?
- Datasets are not introduced apart from the names of the datasets
- Baselines not introduced, unclear which methods are compared and why the comparison matters
- Zero-filling, uniform filling (baselines) not mentioned in the text at all
- The evaluation of uncertainty $\sigma$ could be extended. How well is it calibrated? Does it correlate with low quality imputations? No qualitative evaluations, i.e. exemplary images with high uncertainty. Currently, the only evaluation is the "convergence analysis" in Figure 3.
- The Related Work section partially repeats material from the Introduction (Section 2.1) and the connection between certain subsections and the proposed work could be clarified (Section 2.2). Section 2.3 reads more like a methodological description.

**Detailed Comments:**

- Figure 3: potentially include performance per round as well (should increase)
- The Algorithm 1 seems redundant, all information is in the text

**Justification Of Final Rating:**

The paper presents an interesting approach to incorporating uncertainty quantification into the federated learning pipeline and demonstrates how it can improve overall performance. I believe these insights are valuable to the MIDL community. The rebuttal, particularly the uncertainty evaluation, has substantially strengthened the paper.

**Justification Of The Preliminary Rating:**

The proposed idea is very interesting and the authors demonstrate that adding uncertainty quantification to the federated learning pipeline improves overall performance, however the paper lacks clear structure and does not currently meet the standards necessary for acceptance.

**Questions To Address In The Rebuttal:**

- Address the points raised in the detailed comments
- Clearly introduce baselines
- Add evaluation of uncertainty metric

---

> ### Author Response · Authors · 2026-01-25
>
> We sincerely thank the reviewer for the critical assessment. We fully accept your feedback that the original submission lacked necessary definitions and explicit uncertainty evaluations.
>
> We have used this rebuttal period to perform a major revision of the manuscript, addressing every single point raised in your review. We believe these additions transform the paper into a complete and rigorously validated contribution.
>
> > **Q1: Setting temperature parameter T = 1.0 means no scaling was applied, how was T chosen?**
>
> We apologize for a typographical error in the original manuscript. The experimental value used was T = 0.2, not 1.0.
>
> **Justification:** We chose this low temperature specifically to apply sharp scaling, which down-weights clients with high uncertainty. This ensures that only highly confident imputations significantly influence the global model. We have corrected this value in the revised manuscript.
>
> > **Q2: Datasets are not introduced apart from the names of the datasets**
>
> We apologize for the insufficient dataset description in the original submission. We have now expanded Section 4.1 (Datasets, page 8) to provide comprehensive details.
>
> > **Q3: Baselines not introduced, unclear which methods are compared and why the comparison matters**
>
> Thank you for identifying this gap. We have expanded Section 4.2 (Baselines, page 8) to provide clear descriptions.
>
>
> > **Q4: Zero-filling, uniform filling (baselines) not mentioned in the text at all**
>
> We apologize for this oversight. These baselines are now explicitly described in Section 4.2 (Baselines, page 8) and their results are shown in Table 1(page 9).
>
> > **Q5: The evaluation of uncertainty could be extended. How well is it calibrated? Does it correlate with low quality imputations? No qualitative evaluations.**
>
> We have comprehensively addressed this concern by adding Section 4.5: Uncertainty Calibration Analysis (pages 10-11) with three complementary evaluations:
>
> - **Calibration quality (Figure 4a):** We measure calibration using Expected Calibration Error (ECE). Our reliability diagram shows close alignment between the observed coverage curve and the perfect calibration diagonal, indicating dependable calibration with ECE = 0.0422.
>
> - **Correlation with imputation quality (Figure 4b):** We provide a binned analysis where samples are grouped into deciles by predicted uncertainty. The results show a clear monotonic trend: mean imputation error increases systematically across uncertainty deciles. This confirms that high uncertainty reliably indicates low-quality imputations.
>
> - **Qualitative evaluation (Figure 5):** Side-by-side comparison of low-uncertainty and high-uncertainty chest X-rays, showing that uncertainty meaningfully reflects imputation difficulty.
>
> > **Q6: The Related Work section partially repeats material from the Introduction (Section 2.1) and connection to proposed work could be clarified (Section 2.2). Section 2.3 reads more like methodological description.**
>
> Thank you for this structural feedback. We have restructured the Related Work section in the revised manuscript:
>
> - Eliminated redundancy between Introduction and Section 2.1 by focusing Related Work on technical approaches rather than motivation.
> - Strengthened connections in Section 2.2 by stating how our work addresses the identified gap.
> - Reframed Section 2.3 to function as a comparative literature review rather than a methodological description by introducing alternative calibration approaches (deep ensembles, σ-scaling) alongside β-NLL.
>
> We believe these changes will significantly improve the paper's organization and readability.
>
> > **Q7: Figure 3: potentially include performance per round**
>
> We have implemented this. The revised Figure 3 (right panel, highlighted in red box) now shows AUC progression per round.
>
> > **Q8: Algorithm 1 seems redundant, all information is in the text**
>
> We respectfully choose to retain Algorithm 1 because it serves a distinct purpose from the text: while the text defines the mathematical weighting functions, the algorithm explicitly maps out the procedural control flow (client-side imputation vs. server-side aggregation). We believe this summary significantly aids reproducibility for practitioners implementing the system.

---

> > ### Comment · Reviewer_cxwa · 2026-01-30
> >
> > Thank you for your extensive response to my aforementioned issues. I appreciate the revisions to the text, which I believe have substantially improved its structure and readability. The extended evaluation of uncertainty has further strengthened the paper.

---

### Official Review · Reviewer_ccry · 2026-01-08

**Confidence:** 4
**Preliminary Rating:** 5
**Final Rating:** 5

**Summary:**

The paper "Probabilistic Feature Imputation and Uncertainty-Aware Multimodal Federated Aggregation" introduces a framework designed to handle cases in medical federated learning where some clinical sites are missing entire types of data (modalities), such as text reports for X-ray images. Instead of predicting a single fixed value for missing data, this network outputs a distribution (a mean and a variance). The variance represents the uncertainty of the imputation.

**Strengths:**

The paper is clearly written and shows a substantial contribution. The authors performed a thorough evaluation with reasonable baselines and were able to show that leveraging the uncertainty in the imputed values helps improving classification performance. Moreover, the uncertainty evaluation revealed that increasing communication rounds systematically decreases uncertainty showing that the uncertainty is behaving as desired.

**Weaknesses:**

In general I really enjoyed reading the paper. However, the paper could be improved by extending the evaluation of the probabilistic imputation (i.e. different approaches to model the distribution of the missing values, for example sampling-based approaches). Also the spacing is a bit awkward between the paragraphs in general.

**Detailed Comments:**

Please find my comments above.

**Justification Of Final Rating:**

I am staying with my original evaluation of the paper and think it would be nice to see it at the conference. The authors have provided a satisfactory answer to my concerns about a comparison on additional probabilistic baselines (e.g. sampling is too inefficient in a federated setting) which in my opinion does not make the contribution less significant though.

**Justification Of The Preliminary Rating:**

I think that the paper addresses a nice gap in federated learning and clearly shows how uncertainty can be used to address the issue of missing values. I think it would be very beneficial for the MIDL community.

**Questions To Address In The Rebuttal:**

The manuscript could be further improved by adjusting the spacing between the paragraphs (seems a bit unusual for the MIDL template). Having a few more approaches for probabilistically imputing missing values would make it a bit more stronger but not necessary for conveying the message.

---

> ### Author Response · Authors · 2026-01-25
>
> We sincerely appreciate your positive evaluation and strong support of our work, as well as your constructive feedback.
>
> **On Probabilistic Imputation Approaches:** We acknowledge the suggestion to extend the evaluation to include alternative probabilistic methods, such as sampling-based approaches. While we agree this would offer an interesting comparison, our current design choice which is  a parametric approach optimized via β-NLL was deliberately prioritized to ensure computational and inference efficiency, which is a critical constraint in Federated Learning. Sampling-based methods, while powerful, typically require multiple forward passes or iterative generation steps that burden resource-constrained clients. By utilizing a parametric design, we aim to deliver the necessary probabilistic insights within a single-pass inference budget, which we consider the optimal trade-off for this application.
>
> **On Formatting:** Thank you for pointing the irregular paragraph spacing in the initial submission. We will strictly adhere to the official MIDL template guidelines to correct the paragraph spacing and layout in the final camera-ready version.

---

> > ### Comment · Reviewer_ccry · 2026-01-27
> >
> > Your explanation makes sense to me and my suggestion on additional probabilistic methods was only something that could have made the paper even stronger but not necessary in this context. Good luck with your submission!

---

### Official Review · Reviewer_e2Fm · 2026-01-16

**Confidence:** 4
**Preliminary Rating:** 4
**Final Rating:** 4

**Summary:**

This paper addresses missing-modality challenges in multimodal federated learning for medical imaging by proposing a probabilistic feature imputation network that outputs both mean and per-dimension variance for the imputed modality, trained with β-NLL to avoid variance inflation. The authors leverage these uncertainty estimates locally via a gating mechanism that attenuates unreliable imputed features and globally via a new aggregation rule that upweights clients with more reliable imputations. On federated chest X-ray classification with strong modality imbalance, the approach outperforms deterministic imputation baselines.

**Strengths:**

1. Introduces a probabilistic feature imputation module for federated multimodal learning that directly estimates per-dimension heteroscedastic uncertainty, a gap in prior imputation-based MMFL works.
2. The overall problem setting, method components, and training stages are clearly described, with helpful architectural schematics and a concise algorithm for aggregation.

**Weaknesses:**

1. The paper claims to produce calibrated uncertainty estimates, yet provides no explicit evaluation of calibration quality, such as reliability diagrams.
2. The unimodal and multimodal clients are drawn from different datasets (e.g., CheXpert vs. NIH Open-I), which introduces potential domain shift confounds beyond modality missingness.

**Detailed Comments:**

The paper repeatedly emphasizes that P-FIN produces calibrated uncertainty estimates and leverages these estimates for both local feature gating and global federated aggregation. However, the experimental evaluation focuses almost exclusively on classification AUC, without providing any direct assessment of uncertainty calibration quality.

**Justification Of Final Rating:**

The newly added calibration evaluations and error–uncertainty correlation results strengthen the empirical support for the uncertainty modeling. The clarifications improve the paper, although a more comprehensive assessment under distribution shift would further solidify the safety-related claims. I maintain my original assessment.

**Justification Of The Preliminary Rating:**

I assign a Weak Accept to this paper. The work addresses an important problem in multimodal federated learning for healthcare and proposes a technically sound probabilistic imputation framework that consistently outperforms deterministic baselines, particularly under severe modality imbalance. However, the evaluation lacks explicit uncertainty calibration analysis, and the experimental design conflates modality heterogeneity with dataset-induced domain shift. While these issues weaken the strength of the claims, they appear addressable with additional analysis and clarification.

**Questions To Address In The Rebuttal:**

1. Can the authors provide explicit calibration evaluations of the predicted uncertainty?
2. Have the authors considered or evaluated a setting where unimodal and multimodal clients are constructed from the same underlying dataset?

---

> ### Author Response · Authors · 2026-01-25
>
> We sincerely appreciate the reviewer’s thoughtful evaluation and constructive feedback. Your review correctly identified two key areas where our manuscript needed strengthening: empirical verification of calibration and clarification of the experimental design. We have addressed these points as follows:
>
> > **Q1: Can the authors provide explicit calibration evaluations of the predicted uncertainty?**
>
> Yes, we have now added comprehensive uncertainty calibration analysis in the revised manuscript (Section 4.5, pages 10-11). Specifically:
>
> - **Reliability Diagram (Figure 4a):** We measure calibration using Expected Calibration Error (ECE). Our reliability diagram shows close alignment between the observed coverage curve and the perfect calibration diagonal, indicating dependable calibration with ECE = 0.0422.
>
> - **Error Correlation Analysis (Figure 4b):** We provide a binned analysis where samples are grouped into deciles by predicted uncertainty. The results show a clear monotonic trend: mean imputation error increases systematically across uncertainty deciles. This confirms that high uncertainty reliably indicates low-quality imputations.
>
> - **Qualitative Examples (Figure 5):** We include visual examples showing that low-uncertainty cases correspond to clear imaging with high-quality imputations (top row), while high-uncertainty cases correspond to complex or ambiguous presentations with lower imputation quality (bottom row).
>
> > **Q2: Have the authors considered or evaluated a setting where unimodal and multimodal clients are constructed from the same underlying dataset?**
>
> Thank you for this important suggestion. We acknowledge that our current experimental design (CheXpert for unimodal clients, NIH Open-I for multimodal clients) introduces potential domain shift beyond modality missingness. However, we deliberately chose this setup for two critical reasons:
>
> 1. **Adherence to Standard Benchmarks:**  We strictly followed the experimental protocol established by the baseline **(FIN + FedAvg)**, which also sources unimodal clients from CheXpert and multimodal clients from NIH Open-I. Adopting this identical configuration was necessary to ensure a fair comparison. Deviating to a single-dataset setup would have prevented us from directly benchmarking against the current SOTA under their established conditions.
>
> 2. **Realistic Non-IID Simulation:** Beyond benchmarking, this setup faithfully simulates the structural heterogeneity of real-world federated networks. In practice, sites with different modality profiles—such as specialized imaging centers versus general hospitals inevitably possess different scanner protocols, patient demographics, and disease prevalences. By maintaining this domain shift, we rigorously test whether P-FIN can disentangle "modality missingness" from "institutional bias," evaluating its robustness in realistic, operationally diverse environments.

---

> > ### Comment · Reviewer_e2Fm · 2026-01-28
> >
> > Thanks for the detailed explanation. The newly added calibration evaluations and error–uncertainty correlation results strengthen the empirical support for the uncertainty modeling. The clarifications improve the paper, although a more comprehensive assessment under distribution shift would further solidify the safety-related claims. I maintain my original assessment. Good luck to your submission!

---

### Author Rebuttal · Authors · 2026-01-25

**Rebuttal:**

We sincerely thank all reviewers for their thoughtful and constructive feedback. We have carefully addressed all concerns, particularly the requests for explicit calibration evaluation and clearer presentation. Below, we have uploaded the updated manuscript with the changes highlighted as red.

Summary of Key Revisions:

* **[NEW]** Figure 4: Comprehensive uncertainty calibration analysis
* **[NEW]** Figure 5: Qualitative examples showing uncertainty estimates on chest X-rays
* **[NEW]** Figure 3 (Right): AUC progression per communication round
* **[REVISED]** Expanded dataset descriptions and baseline introductions
* **[REVISED]** Restructured Related Work section for clarity

**Supporting Material:**

/attachment/68e0ee92a01f9ae79de9986f4106c4379b995954.pdf

---

### Meta-Review · Area_Chair_refp · 2026-02-09

**Recommendation:** Accept (Poster)
**Confidence:** 4

**Metareview:**

The problem of missing modalities in federated learning is extremely relevant for the community. The reviewers acknowledge that the proposed approach fills a gap in the literature. The methods are clearly presented. Main points raised by reviewers, particularly regarding uncertainty evaluation, were effectively addressed during the rebuttal. All reviewers recommend acceptance.

---

### Decision · Program_Chairs · 2026-02-13

Accept (Poster)